# Dietary Options for Rodents in the Study of Obesity

**DOI:** 10.3390/nu12113234

**Published:** 2020-10-22

**Authors:** Marianela Bastías-Pérez, Dolors Serra, Laura Herrero

**Affiliations:** 1Department of Biochemistry and Physiology, School of Pharmacy and Food Sciences, Institut de Biomedicina de la Universitat de Barcelona (IBUB), Universitat de Barcelona, E-08028 Barcelona, Spain; mbastipe7@alumnes.ub.edu (M.B.-P.); dserra@ub.edu (D.S.); 2Centro de Investigación Biomédica en Red de Fisiopatología de la Obesidad y la Nutrición (CIBEROBN), Instituto de Salud Carlos III, E-28029 Madrid, Spain

**Keywords:** obesity, diet, high-fat diet, diet-induced thermogenesis, body mass index, rodents

## Abstract

Obesity and its associated metabolic diseases are currently a priority research area. The increase in global prevalence at different ages is having an enormous economic and health impact. Genetic and environmental factors play a crucial role in the development of obesity, and diet is one of the main factors that contributes directly to the obesogenic phenotype. Scientific evidence has shown that increased fat intake is associated with the increase in body weight that triggers obesity. Rodent animal models have been extremely useful in the study of obesity since weight gain can easily be induced with a high-fat diet. Here, we review the dietary patterns and physiological mechanisms involved in the dynamics of energy balance. We report the main dietary options for the study of obesity and the variables to consider in the use of a high-fat diet, and assess the progression of obesity and diet-induced thermogenesis.

## 1. Introduction

Overweight and obesity have increased alarmingly in recent decades. Current public policies and interventions have failed to stop their global prevalence [1]. We know that obesity occurs when there is an imbalance between energy intake and expenditure. A positive energy balance triggers the spill over of lipids that accumulate particularly in adipocytes, increasing their number (hyperplasia) and size (hypertrophy) [2,3]. Both genetic and environmental factors play a crucial role in the development of obesity, and diet is undoubtedly one of the main environmental factors involved. Dietary fat content is often considered the main factor responsible for the increase in adiposity. Human studies have shown that high-fat diets (HFDs) with more than 30% total daily energy intake from fat can easily induce obesity [4,5,6,7]. Other epidemiological studies have shown that the average amount of fat is positively correlated with the incidence of obesity [8,9]. This has led public policies to promote a decrease in the amount of fat in the human diet.

HFDs induce obesity not only in humans, but also in other species [10]. For example, in rats [11] and mice [12], the level of dietary fat has been related directly to an increase in body weight and body fat [13,14]. Animal models have been widely used in food and obesity experiments [15,16]. In these experiments, a variety of commercial diets have been used to induce obesity, such as HFDs ranging from 30% to 78% of total energy intake from fat [17]. Animal models are useful for studying obesity, as rodents easily gain weight when fed an HFD [18]. Other factors might directly contribute to diet-induced obesity, including the profile of fatty acids in the diet [19], an increase in adipose tissue lipoprotein lipase activity [20], an increase in the amount and decrease in the frequency of food consumption [21], excessive energy consumption attributed to the high energy density of the diet [22,23], the sensory and organoleptic fat characteristics, and low dietary satiety [22].

To effectively prevent and treat obesity, we must better understand energy homeostasis in the body and rethink why obesity is so resistant to treatment [24]. The objective of this review is to report the factors that can play a fundamental role in the development of obesity and explore the mechanisms involved in the development of HFD-induced obesity, the fine-tuned equilibrium in energy dynamics, the importance of diet-induced thermogenesis, the use of rodents as animal models of obesity, and their implication for the treatment of human obesity in terms of changes in body composition.

## 2. Animal Models of Diet-Induced Obesity

Rodent animal models have become very useful for the study of obesity. One of the advantages is that the diet given to the entire sample group can be standardized in terms of the percentage of macronutrient distribution. However, although most rodents tend to become obese under HFDs, there might be variability in body weight gain, glucose tolerance, insulin resistance, blood lipid content, and other strain-dependent parameters [25].

An interesting study carried out in nine strains of inbred mice (AKR/J, C57BL/J, A/J, C3H/HeJ, DBA/2J, C57BL/6J, SJL/J, I/STN, and SWR/J) has evaluated dietary obesity on body composition and energy intake [19]. The authors observed that only six mouse strains showed an HFD-induced increase in adiposity, while in the SJL/J, I/STN, and SWR/J strains no significant changes were observed. Moreover, only the AKR/J mouse strain showed higher energy intake [19]. This highlights the importance of the mouse strain selection in the study of diet-induced obesity.

Some mouse strains, such as C57BL/6J or AKR/J, are more susceptible to the development of obesity when they are fed an HFD [26]. Under an HFD, these two mouse strains have similar degrees of weight gain. However, C57BL/6J mice are more glucose intolerant than AKR/J mice [26]. Therefore, C57BL/6J mice are preferred to induce obesity with diet. Since C57BL/6 mouse strains are widely used in diet-induced obesity (DIO) studies, different substrains of C57BL/6 mice are also available. They include genetic and environmental differences such as those present in the microbiome [27] as well as differences in response to an HFD feeding [27,28,29,30]. A study carried out in C57BL/6JRj and C57BL/6NTac mice substrains, reported important differences in response to an HFD, and concluded that the C57BL/6JRj strain is protected against DIO regardless of physical activity and food intake [29]. Another study carried out in C57BL/6N and C57BL/6J mice substrains showed key genetic differences, in different loci of single nucleotide polymorphism, associated with different responses to DIO [30]. Differences in glucose homeostasis and insulin sensitivity were also observed [30].

In the case of rats, the Sprague–Dawley and Wistar strains are very popular as animal models for studying obesity as they easily gain weight with an HFD. However, these strains are known to have a variable weight gain response [31]. Some animals gain weight rapidly, while others gain the same as rats fed a low-fat diet [32]. Furthermore, body weight gain in these rat strains is often similar under a 45% or a 60% kcal from an HFD. This should be considered in the diet selection process for these strains [33] (see Figure 1A). However, this diversity could be interesting, since it might represent what happens in the development of human obesity.

Rodent models have been an essential tool in the study of obesity. They are a convenient model for drug development and new dietary interventions [34]. However, when comparing with humans there are important limitations that must be considered. A recently published systematic review evidences limitations in obesity studies [35]. These limitations include that the induction of diet in rodent animal models are long-term studies, difficult to standardize and in many cases the obese phenotype presents a metabolic exaggerated response for the HFD. In addition, the different physiological characteristics between rodents and humans [35] must be taken into consideration. The main limitation of obesity studies with rodents is the dietary applicability in humans. In animal studies, both the exact diet composition and the consumption amount per rodent could be controlled. However, in humans only dietary consumption could be estimated. In addition, in humans weight gain and loss will depend on the great individual and genetic variability [35].

Overall, when an obesity induction study is planned, the animal strain and substrain, the composition of the diet and the limitations when translating the results into humans must be carefully considered. It is also important to make a good dietary choice to efficiently induce obesity in the selected animal model.

## 3. Assessment of Diet-Induced Obesity

Obesity might be evaluated in rodents in the same way as in humans, using criteria based on weight gain [36] and increased body fat content [37]. However, obesity ranges such as the body mass index (BMI) in humans have not been classified for rodents. In most rodent studies, the degree of DIO is assessed by comparing the body weight of the experimental group with that of the control group, which is normally fed a low-fat diet [38,39,40]. Importantly, comparisons of rodent control groups of the same strain and age, fed with similar chow diets, show variations between 10 and 25% in body weight. These results indicate that there is intrinsic individual variation [39,40]. The term adiposity index is widely used in obesity studies to determine body fat in rodents. It is calculated as the sum of the mesenteric, epididymal, retroperitoneal, and perirenal fat depots’ weights divided by the total body weight (g) × 100 and it is expressed as a percentage of adiposity [41].

The Lee index was defined a few decades ago [42] to classify obesity in rats in a similar approach to the BMI used for human obesity. The Lee index is defined as the cube root of body weight (g) divided by the naso–anal length (mm). Lee index values above 310 g were considered an indicator of obesity in rats. Correlations were also found between the Lee index and body fat content [43,44,45,46]. More recently, researchers have used the Lee index to assess the degree of obesity in rats [47,48,49] and in some cases to estimate the body fat composition in normal and obese mice [46].

In humans, obesity is assessed through anthropometric measurements based on BMI and direct adiposity measurements to diagnose the nutritional status. The relationship between BMI and adiposity has been shown to be influenced directly by factors such as gender and age [50]. To obtain a more precise measurement of the degree of obesity, methods such as air displacement plethysmography, and dual energy X-ray absorptiometry have been used [50,51]. Dual energy X-ray absorptiometry has also been used to assess body composition in rats [38,52], and HFD-treated rats have shown a linear correlation between total fat and body weight [53,54]. However, scientific evidence shows that measuring body fat to assess obesity in animals is an even more sensitive criterion. Researchers found that rats fed a 40% HFD for 10 weeks showed only a 10% increase in body weight. However, their actual total increase in body fat was 35–40% compared to animals fed a low-fat diet [39]. Importantly, animal models of dietary obesity are also classified as prone or resistant to obesity according to criteria such as weight gain, body fat and urine norepinephrine concentrations [55,56,57,58,59].

It would be useful to standardize and validate the criteria used to classify the degree of obesity in experimental animal models. This would help researchers to standardize results and extrapolate them to human obesity studies.

To determine body fat in rodents in the study of obesity, the adiposity index (AI) has been described, which is determined by the sum of the weights of white adipose tissues (Mesenteric, epididymal, retroperitoneal and perirenal fat tissues) divided by body weight (g) × 100 and is expressed as a percentage of adiposity [41].

## 4. Obesogenic Diets

Several diets are available to induce obesity in rodents. The most widely used are known as diet-induced obesity (DIO) models, which consist of diets with varying fat content (i.e., % of kcal from fat), mainly at 10%, 45%, and 60% [25]. Rodents and humans need a balanced, fresh, healthy diet that meets their nutritional needs. The metabolic phenotype resulting from chronic shortage of certain macro- and micronutrients in the diet can vary drastically [60]. Therefore, for experimental purposes, it has been established that a diet considered normal for rodents would contain approximately 10% fat. Thus, in the DIO series, a 10% fat diet would be established for the control group, while a 45% or 60% HFD would be selected for the experimental groups. However, experimental use of these diets in rodents has been questioned recently, since the fat levels are much higher than the fat percentages found in diets normally consumed by humans with obesity [61].

A 60% HFD: From an economic perspective, the use of a 60% HFD is very effective since it rapidly induces the obese phenotype in rodents. It has been shown to trigger a higher degree of obesity and to have a greater impact on glucose metabolism and insulin resistance [12,62]. Therefore, it is more convenient for researchers, since it reduces the length of experiments and animal housing time, which cuts animal facility expenses. However, a 60% HFD distorts the fat content of food to a much greater extent, especially when compared to rodents with a normal diet (10% fat). Thus, it leads to a more exaggerated metabolic response. Because of this, in studies that try to mimic the physiology of humans with obesity, it has been considered less physiological than a 45% HFD [61]. In addition, the percentage of sucrose is higher in a normal 10% fat diet and decreases in a 45% HFD and even more in 60% HFD. To this end, many commercially available additional versions of a normal 10% fat diet try to match the sucrose levels present in 45% and 60% HFDs, which range from 7% to 35% sucrose [61].

A 45% HFD: Diets with 45% fat also induce obesity, but more slowly than a 60% HFD, although this induction is more similar to the development of human obesity. An interesting study compared the metabolome of mice fed a low-fat, high-fat (45%), and very high-fat (60%) diet and revealed that 80 out of a total of 91 analyzed metabolites were altered in HFD vs. normal diet-treated mice [63]. Importantly, only 35 of the 80 modified metabolites were common to the two obesogenic diet groups, which indicates that the diets trigger different metabolic responses [63]. Thus, to induce obesity, it is important to consider variations in the development of obesity, the physiology and the metabolism of rodents fed with different types of HFD.

Cafeteria diet: The above commercially available HFDs are characterized by an inverse relationship between the percentage of fat and sucrose. However, there are other dietary options, such as the “cafeteria diet”. In this model, the animal selects its own food, which is very tasty and easily accessible. The cafeteria diet is mainly composed of foods available in supermarkets that are high in sodium, fat, and sugar, such as biscuits, sweets, cheese, and sausages. This diet has been studied in DIO models in rats [64,65,66,67] and mice [68,69,70]. A major disadvantage of the cafeteria diet is the lack of defined composition, since the animal can choose a varying amount of the items in the diet every day. Furthermore, animals can make different food choices, leading to a lack of uniformity in the food intake. Therefore, this diet is less precise and less reliable in terms of caloric intake for obesity studies [71].

Types of fats: Several studies suggest that the type of fat may play a crucial role in the development of obesity [72]. Not all fats are obesogenic. In other words, the main contributor to the development of obesity is the profile of fatty acids in the diet rather than the energy content of the ingested fats [73,74,75]. This is important in the selection of a type of HFD, since the composition of dietary fatty acids can affect the results [76,77]. Most of the HFDs used in DIO studies contain high amounts of saturated and/or trans fats. As described above, they are very effective in inducing obesity and metabolic diseases in susceptible strains [78,79]. Oils rich in monounsaturated fatty acids, such as n-3 fatty acids, have been characterized by their health benefits and their ability to counteract HFD-induced obesity [80,81,82]. Fatty acids could modify the metabolic phenotype through various mechanisms, such as altering membrane permeability, gut microbiota or the expression of genes involved in energy homeostasis, insulin action, and eicosanoid and cytokine production [83,84,85]. The anti-inflammatory, hypotriglyceridemic and anti-obesogenic effects of long chain omega-3 polyunsaturated fatty acids (n-3 PUFA) are well known. The proposed n-3 PUFA mechanisms may include the modulation of lipid metabolism and adipokines, decreasing adiposity and adipose tissue inflammation, promoting adipogenesis and altering epigenetic mechanisms. However, their role in human diet therapy is still under debate [86].

Thus, in the selection of a suitable diet to study DIO in rodents, it is important to consider the diet nutritional proportions, the composition of the macro- and micronutrients, the types of fatty acids, the real caloric intake and the development of the metabolic phenotype (see Figure 1B).

## 5. Physiological Mechanisms Involved in Diet-Induced Obesity

It is well-known that obesity progression is accompanied by an alteration in several key physiological mechanics. Regarding adipose tissue, it was initially believed that the number of adipocytes was determined in the first years of life and that future obesity developed during adulthood was the result of an increase in adipocyte size [87]. However, hyperplasia is now known to be a continuous event at any stage of life, since when adipocyte hypertrophy occurs, factors such as TNF-α and growth factors that stimulate adipocyte hyperplasia are released [2]. Likewise, studies on the reversal of obesity in humans have found decreases not only in fat cell size but also in adipocyte number [3].

A fundamental physiological mechanism involved in the progression of DIO is the adaptation of the organism to conditions of metabolic stress that could ultimately progress into a pathological situation. In this sense, there are several studies that have investigated the reversal of obesity through dietary interventions [88,89,90]. Cucurbita maxima seed oil (CSO) supplementation in HFD-fed obese rats improved the obese phenotype through significant changes in body weight and lipid metabolism including alterations in LDL, HDL, triglycerides, total cholesterol, adiponectin, and leptin levels [88]. The authors also observed changes in metabolic enzymes such as fatty acid synthase, acetyl-CoA carboxylase, carnitine palmitoyltransferase-1, HMG-CoA reductase, and in inflammatory markers such as TNF-α and IL-6. Another interesting study demonstrated the reversibility of obesity and diabetes in C57BL/6J mice by a reduction of the dietary fat [89]. Thus, this highlights the adaptive capacity of the organism to remodel the metabolic phenotype according to dietary changes.

Here, we focus on hormonal alterations that occur during diet-induced obesity. Adipose tissue, the stomach, pancreas, and other tissues send signals to the brain to regulate energy balance, through the secretion of certain hormones [91]. Hormones can affect energy balance by reducing or increasing energy expenditure and food intake and they are regulated and secreted by the tissue adipose (such as leptin, adiponectin, and resistin), stomach (ghrelin), pancreas (insulin, glucagon), intestine (cholecystokinin, incretins), among others. All of them have a fundamental role in energy balance and food intake [87]. Here, we will describe the main regulatory hormones involved in the metabolism of adipose tissue, stomach, and pancreas, the key organs implicated in the progression of DIO. Thus, next we will focus on three relevant hormones that orchestrate energy balance and play a critical role in obesity: leptin, ghrelin, and insulin. We will also address the role of the sympathetic nervous system as an integrator of the central and peripheral signals in obesity.

Leptin: Leptin plays a key role in the control of food intake and body weight [92]. Leptin is secreted by adipose tissue in an obesogenic situation, and is usually correlated with fat mass. Increased leptin plasma levels result in decreased food intake and increased energy expenditure [92]. Human and rodent studies have shown that obesity is associated with higher plasma leptin concentrations [93]. At the beginning of the HFD treatment, rodents are sensitive to the leptin-induced reduction in food intake. However, despite increased plasma leptin levels, the animals ultimately gain weight as a result of increased feed efficiency and the development of leptin resistance [94]. This implies that, after prolonged HFD feeding, leptin resistance is the consequence of the obese state and not the cause of obesity development. Other studies found that the composition of fatty acids in an HFD could influence circulating leptin levels [95]. These studies concluded that the site of fat accumulation depends on the dietary fatty acid profile, and that various deposits of adipose tissue may contribute differently to leptin circulating levels [96].

Ghrelin: Ghrelin is a peptide secreted by the stomach that stimulates the release of pituitary growth hormone [97]. Circulating levels of ghrelin increase before and fall after each meal ad libitum, which increases food intake [98]. Suggested mechanisms for ghrelin’s actions are hypothalamic stimulation of neuropeptide secretion and decreased fat oxidation and use to initiate the eating process and ultimately increase food intake [98]. HFDs are known to regulate ghrelin’s secretion inversely to leptin levels [99]. Since the suppression of ghrelin levels after a meal is associated with postprandial satiety, decreased suppression of ghrelin secretion after HFDs may explain HFD-induced hyperphagia. Therefore, under an HFD, impaired ghrelin suppression after a meal causes excessive energy consumption and induces obesity. Furthermore, obesity itself affects suppression of ghrelin secretion after a meal, which further exacerbates the development of obesity [2,100].

Insulin: Insulin is a polypeptide hormone that is produced and secreted by the pancreatic beta cells to ultimately lower blood glucose levels. Elevated plasma insulin levels and resistance to the metabolic effects of insulin are associated with obesity progression [2]. Regardless of obesity, HFDs contribute to glucose intolerance and insensitivity to the hypoglycemic effect of insulin [100]. The fatty acid profile of the diet also plays a crucial role in HFD-induced insulin resistance [100]. In fact, a linear relationship has been found between the percentage of intolerance to glucose and the fat content in diet [12,38]. Differences between dietary fatty acids affect the composition of cell membranes. In turn, this influences the affinity of insulin to its insulin receptor, and thus affects insulin action [100]. Some studies have found that insulin secretion and sensitivity positively correlate with the degree of fatty acid unsaturation, especially with an n-3 diet. Saturated fatty acids result in more insulin resistance [100]. It has been shown that excessive consumption of saturated fatty acids (SFA) promotes lipid storage in liver and visceral fat both in humans and rodents, while energy excess from polyunsaturated fatty acids (PUFAs) might promote lean mass [101,102,103]. In fact, it has been demonstrated that n-26 PUFAs, compared to SFA, are able to reduce liver fat and to improve the metabolic status, without changes in body weight, oxidative stress, and inflammation [102]. However, in terms of inducing insulin resistance, the effects of SFA vs. PUFA might not necessarily differ [103]. In a study conducted in mice fed obesogenic diets with different fatty acids composition, it has been shown that regardless of the fatty acids’ composition (SFA or PUFA), the HFD was able to induce similar weight gain and insulin resistance [103]. Importantly, an SFA-rich diet containing docosahexaenoic acid (DHA) and eicosapentaenoic acid (EPA) efficiently reduced hepatic steatosis and improved insulin sensitivity [103]. Thus, clear differences are seen in the obese metabolic phenotype depending on the fatty acid composition of the diet. Apparently, these differential effects of various dietary fatty acids might also depend on the total amount of lipids present in the diet [104].

The sympathetic nervous system: The brain is an important player in obesity’s development. It receives nutrients (such as glucose and lipids) and hormonal signals (such as leptin, insulin, ghrelin, and thyroid and gonadal hormones) that are mainly integrated in the brain region of the hypothalamus. This region is organized in multiple clusters of neurons acting as a network and sending signals outward via the sympathetic nervous system which directly contacts different tissues and organs to closely control their metabolism and whole-body energy homeostasis.

Compelling evidence underlines that sympathetic neurons innervate liver, pancreas, gut, and adipose tissues with a different distribution of nerves and a bi-directional communication between brain and these peripheral tissues. Here, we will focus on the role of the sympathetic nervous system in the fat depots. In brown adipose tissue (BAT), the secretion of norepinephrine at the terminal nerves that surround adipocytes activates the β3-adrenergic receptor and signaling pathways such as lipolysis, fatty acid oxidation, and thermogenesis [105,106]. Surgical or chemical denervation of BAT decreases UCP1 expression and highlights the functional relevance of BAT innervation in regulating thermogenesis, energy expenditure and body mass [107,108,109]. In addition, it has been suggested that alterations in the crosstalk between BAT and brain may activate the sympathetic innervation of other adipose tissues to modulate energy balance. Nguyen et al. demonstrated that BAT sympathetic denervation is able to activate the sympathetic activity of other adipose tissues promoting browning in inguinal white adipose tissue (WAT) [109]. WAT depots are also innervated by the sympathetic neurons and β3-adrenergic signaling is necessary to activate lipolysis [110,111]. In addition, WAT sympathetic activation produces different effects depending on the specific fat depot. Under sympathetic activation subcutaneous WAT showed a reduced expandability ability than visceral WAT, although both tissues showed reduced adipocyte proliferation [112]. These results suggest a different fat depot pattern of sympathetic innervation which has been recently confirmed by using new microscopy methods for whole-depot nerve imaging [113].

Diet can modulate sympathetic activity in adipose tissues. Compelling evidence suggests that in conditions of calorie excess such as under an obesogenic diet the BAT sympathetic activity increases to promote energy expenditure. Levin et al. showed that acute HFD feeding increases the norepinephrine turnover in BAT but that it decreases after long HFD treatment [114]. More recently, it has been shown that HFD increases UCP1 thermogenic protein in mice and rats [115,116] and BAT temperature after 20 weeks of diet [117]. Importantly, it has been demonstrated that the levels of gamma-aminobutyric acid (GABA), an inhibitory neurotransmitter that suppresses neuron excitability, are increased in BAT of HFD-fed mice promoting BAT disfunction [118]. All these exiting results provide new insights into innervation patterns and nerve plasticity in adipose depots, suggesting that neuromodulation could become an essential strategy to explore new ways to treat obesity (reviewed in [119,120,121,122]). Additional research should be done to elucidate nerve subtypes and their tissue functions in order to understand the physiological crosstalk between brain and peripheral tissues and their contribution to maintain energy balance.

The hormonal behavior during obesity progression is quite similar in humans and rodents. Thus, rodent models could be useful to study the effects of dietary fatty acid profile on the endocrine system, the hormonal pathway of action under the effect of an HFD in obesity progression and the effect on energy consumption and/or expenditure. Rodent models could also be used to study energy dynamics from diet-induced thermogenesis and its metabolic pathways in the induction of obesity through diet.

## 6. Other Factors Involved in Obesity Progression

Several factors are involved in DIO progression, including behavioral mechanisms. One is sensory facilitation of intake due to the intrinsic sensory characteristics of fats. Sensory stimulation of food consumption could directly influence energy intake [123], promoting digestion, absorption and, in some cases, consumption selection [124], thus contributing to increased diet-induced thermogenesis [125]. Sensory properties contribute to the high palatability of HFDs. Texture and odor have a particular impact and predispose to a sensory preference in animal models [22]. Previous studies suggested that the satiety signal of fats is weaker than that of carbohydrates and proteins, since fats play a fundamental role in the excessive energy consumption of HFDs [126]. Low energy diets trigger more stomach bloating than diets with higher energy density, and therefore induce a feeling of satiety that is greater than that caused by fats [6]. A study with rats concluded that the post-ingestion effects of fats increase daily caloric intake by triggering an increase in food intake (hyperphagia). During a meal, fats suppress caloric intake less than carbohydrates [127], and this post-ingestion effect could also increase caloric intake by conditioning sensory preference [128].

Other factors are the rhythmicity of the feeding process and stress, which could contribute directly to obesity progression. Rhythmicity has been shown to play a critical role in the development of obesity [129]. Unlike humans, rodents are nocturnal animals that eat 70 to 80% of their food during the dark phase. They eat at the beginning of the night and at the end, that is, at dusk and dawn [130]. Stress is another factor in the behavioral mechanisms of HFD-induced obesity. Previous studies have shown that long-term stress increases food intake and promotes weight and fat gain in humans [131]. However, a different pattern of stress response has been demonstrated in some rodent models [59]. For example, obesity-prone and obesity-resistant rats have been found to vary in their ability to respond to stress [132]. The effect of an HFD on weight gain after stress was studied and it was observed that stressed rats lost weight regardless of diet. In other words, low fat diets and HFDs produced similar changes in body weight under severe stress. However, with mild stress, the high-fat fed rats responded better to the effect of stress weight reduction [133]. In mice, a greater preference for an HFD has been reported during chronic stress [134]. However, the mechanisms that influence food intake during acute and chronic stress vary [135].

In rodents, total energy expenditure is the sum of basal metabolic rate (BMR), physical activity, diet-induced thermogenesis (DIT), and cold-induced thermogenesis (facultative or adaptive) [136]. Currently, most of the metabolic and obesity studies are carried out at room temperature of 21 °C, which is considered the thermoneutral zone for adult humans. However, mice subjected to the same temperatures experience a chronic cold, because their thermoneutral zone is 30 °C [137]. It has been reported that mice show differences in metabolic phenotype when housed at room temperature (Ta) (21 °C) vs. at thermoneutrality (30 °C). Chronic cold triggers controlled hypothermia and energy expenditure is affected by changes at the physiological (food intake, adipose tissue physiology, and increased adaptive thermogenesis) and metabolic (such as basal metabolism, adaptive thermogenesis, dietary efficiency, insulin secretion, adipose tissue physiology, adipocyte, and vascular inflammation) levels [137]. During exposure to chronic cold (21 °C), total energy expenditure may increase by more than one third, due to the cold-induced thermogenesis required to maintain basal temperature [138]. Because energy expenditure decreases by approximately 50% in mice housed at thermoneutrality, the metabolic phenotype in obesity, including adiposity, is highly dependent on Ta [139]. These suggest that chronic housing of mice under temperature stress conditions may mask the genetic functions involved in energy balance and metabolic homeostasis by causing a change in the metabolic phenotype [140]. Some studies have shown that the lack of efficient induction of the obese phenotype by an HFD is due to the fact that the mice were housed in conditions below their thermoneutrality [105,141,142,143], demonstrating the correlation between low temperature and low effectiveness of the diet, as well as altered energy homeostasis [144,145].

Thus, in addition to dietary factors and the animal model under study, the other factors that could contribute to DIO progression must be monitored, such as the sensory properties, rhythmicity in the feeding process and the temperature stress to which the study model is exposed. All of these factors could directly contribute to the efficiency of the DIO, and somehow bias the experimental results.

## 7. Diet-Induced Thermogenesis

Diet-induced thermogenesis (DIT), also known as the thermic effect of food or the specific dynamic action of food, is defined as the increase in energy expenditure after food consumption. It is expressed as a percentage of the increase in energy expenditure above the reference level due to the energy content of the food eaten and it is usually expressed as a percentage [146]. Total energy expenditure (TEE) is comprised of the contribution of resting energy expenditure or the basal metabolic rate (BMR), the energy required for physical activity and DIT. Although DIT is the smallest component of the three, it could play a fundamental role in the development and/or maintenance of obesity [146] (See Figure 2A).

## 8. Components of DIT and Energy Expenditure

The increase in energy expenditure associated with DIT has two main components: one mandatory that is required for the digestion, transport, and storage of nutrients, and one optional for heat production [147]. Most of the energy (30%) is required for the storage of nutrients as glycogen and triglycerides [148]. The proportion of energy required for digestion and transport is substantially less than the energy required for nutrient storage [149]. Thus, the food-related increase in energy expenditure is primarily the sum of heat and energy storage. Heat loss represents part of the energy produced in the immediate hours after eating. DIT could increase energy expenditure depending on the amount and composition of ingested macronutrients. DIT reaches the maximum peak of thermogenesis between 1 and 2 h after ingestion. Then, it decreases, and the body returns to the BMR (see Figure 2B). The representative average of total energy expenditure corresponds to 60–70% of BMR, 10–20% of physical activity, and 5–15% of DIT [150].

In rodents, total energy expenditure is the sum of BMR, physical activity, food thermogenesis, and cold-induced thermogenesis (facultative or adaptive) [151]. During exposure to chronic cold (21 °C), total energy expenditure may increase by more than a third, due to the cold-induced thermogenesis required to maintain basal temperature [137]. Energy homeostasis is strictly regulated by hormones that have a fundamental role in the regulation of thermogenesis [152]. In living organisms, the chemical energy of food is primarily converted to heat through complex biochemical processes of energy metabolism. BAT controls thermogenesis by uncoupling oxidative phosphorylation from ATP production, which leads to the generation of heat. This process is tightly related to important metabolic pathways such as lipolysis, lipogenesis, mitochondrial respiration, or fatty acid uptake, transport, storage, and oxidation [150,153].

## 9. DIT Dynamics in Obesity

Decades ago, the possibility was first raised that differences in the thermogenic response to food might contribute to the development of obesity [154]. However, many researchers consider that there is not enough evidence to support the theory of an altered DIT in obese individuals [155]. Other factors have been associated with the pathogenic role of thermogenesis that may greatly contribute to the development of obesity [156]. Changes in caloric intake alter the energy balance, which contributes to the dynamics of total energy expenditure. The static or linear energy balance theory assumes that a change in energy intake does not change or influence energy expenditure. However, the dynamic or non-linear energy balance theory assumes that numerous biological and behavioral factors regulate and influence the energy balance [24]. In the theory of dynamic energy balance, which is most studied today, it is considered that a change in factors related to energy intake may influence factors related to energy expenditure. Thus, reducing or increasing caloric intake could change all aspects of energy expenditure, including resting metabolic rate, exercise metabolic rate and adaptive thermogenesis. A large body of evidence shows that the metabolically active component, which refers to fat-free mass, decreases or increases this energy expenditure [24].

Previous studies have shown that the maintenance of either a low or a high body weight is associated with compensatory changes in energy expenditure [157]. On the contrary, when trying to achieve a different body weight than usual, these compensatory changes may explain the poor long-term efficacy of anti-obesity dietary therapies [157]. Importantly, in humans’ individual differences rather than group means should be further explored to identify the nature of these specific “compensators” and “non-compensators “of the energy balance [158].

Thus, more research is needed to identify all the components that contribute to the energy balance regulation. For this purpose, the rodent animal model where the different components of the energy balance could be tightly controlled might be useful.

## 10. Food Efficiency and DIT

Some studies have attributed HFD-induced obesity to the high dietary efficiency of fats. Lipids have a higher energy content and, therefore, a greater effect on body weight increase than other macronutrients such as carbohydrates and proteins [5]. The biochemical processes involved in nutrient digestion, absorption, and storage are associated with energy consumption in the body that is equivalent to 25 to 30% for proteins, 6 to 8% for carbohydrates, and 2 to 3% for fats. This energy is correlated with food efficiency according to the macronutrients’ metabolic fate. The metabolic efficiency is 92 to 94% for carbohydrates, 70 to 75% for proteins, and 97 to 98% for fats [4].

The high energy density of HFDs also contributes to their greater food efficiency. In humans, after two weeks of exposure to ad libitum diets, we learn to compensate for the higher energy density of the diet and eat less food based on our energy requirements [159]. Rodents have been described as hyperphagic when they are fed an HFD, since they ingest more energy than they require [39,55]. Although studies do not always report the amount of diet delivered, rats have been shown to attempt to adjust their food intake according to the energy density of the HFD [160]. However, researchers revealed that rats were unable to fully adjust to the additional dietary energy, despite ingesting a minimal amount of the HFD to meet their minimum requirements (for maintenance and growth) [114].

Overall, it is important to assess the caloric distribution that is delivered to the study model, considering the energy expenditure related to the contribution of nutrients and their metabolic demand. Furthermore, the HFD used to induce obesity must comply with the minimum requirements of macronutrients and micronutrients required to carry out metabolic processes. Thus, we should consider the dynamics of the energy balance and the changes in DIT. All of these factors are important to interpret the hyperphagia derived from an HFD and the overall scientific results obtained from the DIO model.

## 11. Conclusions

Obesity can be caused by physiological and behavioral mechanisms since both contribute to the development of the obese phenotype. The physiological mechanisms involved in HFD-induced obesity are mainly due to excessive energy consumption and a low satiating effect because of the high energy density of fats.

However, behavioral mechanisms could also contribute to the development of obesity. These include sensory facilitation of fat intake that is also regulated by the rhythmicity of food intake. Due to the high energy intake, the caloric intake from fat largely contributes to DIT. Currently, there is a lack of evidence that directly links body weight to the fatty acid profile of HFDs. This would be an interesting field of study. A key point in the design of animal studies is that HFDs must meet the minimum nutrient requirements, especially for proteins, vitamins, and minerals. Excessive HFD consumption has a high energy contribution but would not meet the minimum micronutrient needs (vitamins and minerals).

Overall, for correct DIO experimental design it is important to consider various controllable factors such as the benefits and limitations of the selected animal model, and the appropriate dietary choice considering its components and caloric distribution. Thus, in the study of obesity with rodent animal models, we strongly recommend on the publication of full details including age, strain, substrain, and supplier of the animals used, caloric intake, and energy density, macro and micronutrients, and fatty acid composition, and supplier of the diets. All of this information will extremely help the research community to correlate the metabolic phenotype with the efficiency of the diet’s obesity-induction.

## Figures and Tables

**Figure 1 nutrients-12-03234-f001:**
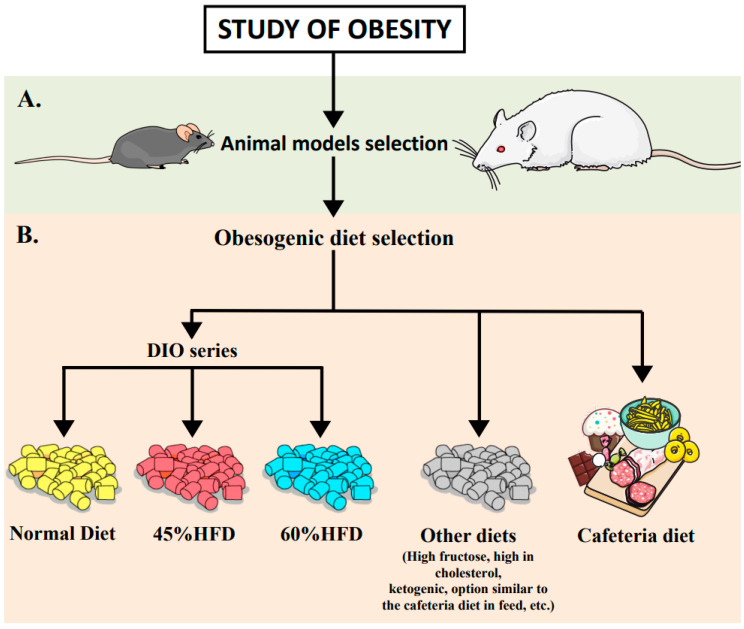
Selection of the animal model and diet for the study of obesity. (**A**) Selection of the animal model. Important factors to consider in the selection of the animal model to study obesity include the animal species, strain and the analysis of physiological mechanisms involved in obesity progression such as adiposity, hormonal regulation, and stress- or diet-induced thermogenesis (DIT) dynamics. (**B**) Selection of the obesogenic diet. The obesogenic diet might be selected according to the objective of the diet-induced obesity (DIO) study, specifying the best diet for the control and experimental groups and considering the caloric distribution of macronutrients, micronutrients, and the types of fat content, and comprehensively monitoring the animal model in the DIO study.

**Figure 2 nutrients-12-03234-f002:**
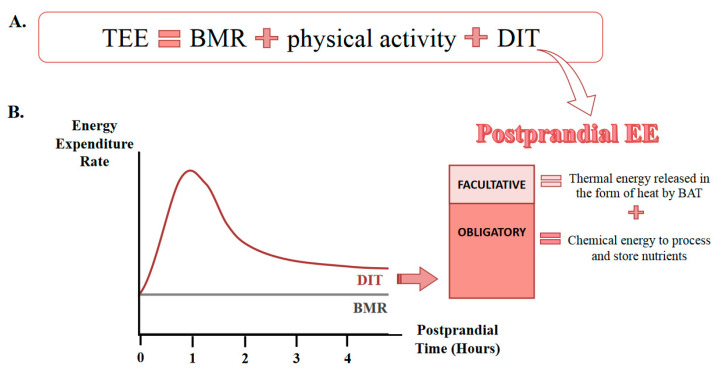
Representation of the components of total energy expenditure (TEE) and the increase in postprandial energy expenditure. (**A**) The components of TEE are the basal metabolic rate (BMR), the energy expenditure involved in physical activity and diet-induced thermogenesis (DIT). (**B**) DIT presents a thermogenic postprandial peak and consists of two parts: an obligatory component of the energy required for the storage, processing and transport of nutrients; and a facultative component of the thermal energy generated by brown adipose tissue (BAT). The energy required for nutrient storage comprises most of the mandatory component.

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
