# Peer review of "Dietary Options for Rodents in the Study of Obesity"

_nutrients, 2020, doi:10.3390/nu12113234_

Round 1
Reviewer 1 Report
The manuscript by Bastías-Pérez et al. aims to review the current state of knowledge about nutritional strategies that can be used to induce obesity in laboratory rodents. Obviously, given the growing prevalence of obesity worldwide, this is therefore a very topical issue. However, in its present form, the manuscript does not represent major contribution to the field. To become a reliable a condensed form of information, it needs to be improved in various places, both with regard to (i) the completeness of the available scientific knowledge and the extension of certain topics and (ii) the comprehensibility and accuracy of the text.
(i) completeness of the available scientific knowledge and the extension of certain topics
Section 2. Animal models of diet-induced obesity
- include and comment on some classic works on this subject, e.g. West DB et al., Am J Physiol 1992.
- with regard to the widespread use of inbred C57BL/6 mice for the study of diet-induced obesity, it is necessary to present and comment on other publications that address the differences specifically between different substrains of C57BL/6 mice (e.g. C57BL/6J vs. C57BL/6N, and possibly others), mainly with respect to the induction of obesity and glucose homeostasis disorders.
Section 3. Assessment of diet-induced obesity
- include the term „adiposity index“ and comment on its applicability in the study of diet-induced obesity in rodents
Section 4. Obesogenic diets – Types of fats
- …n-3 fatty acids, have been characterized by their health benefits….it is necessary to inform the reader that dietary n-3 fatty acids counteract HFD-induced obesity in rodent models before mentioning various types of the mechanisms behing the effects of these lipids.
Section 5. Physiological mechanisms involved in diet-induced obesity
- include the important topic of „adaptation“ vs. „pathology“ when discussing the effect of high-fat feeding on the development of obesity and metabolic alterations in rodent models (in this regard, it may be valuable to discuss studies involving the reversal of obesity and/or metabolic changes due to the transition to standard feed in animals that have previously been fed a high-fat diet)
- add some explanation for why you chose those three hormones (i.e. leptin, ghrelin, insulin)
- elaborate further the topic focusing on fatty acid composition of the diet and its effect on fat accumulation and insulin sensitivity; there seems to be a good evidence of a differential effect of saturated vs. polyunsaturated fatty acids of n-6 series on fat accumulation in various depots of adipose tissue as well as in the liver of both humans and rodents (e.g. Rosqvist F et al., Diabetes 2014; Bjermo H et al., Am. J. Clin. Nutr. 2012; Pavlisova J et al., Biochimie 2016), while in terms of inducing insulin resistance, their effects may not necessarily differ (e.g. Pavlisova J et al., Biochimie 2016). Apparently, the differential effects of various dietary fatty acids may also depend on the overall amount of lipids in the diet (e.g. Vessby B et al., Diabetologia 2001).
- the role of the activity of the sympathetic nervous system should be mentioned and discussed indeep
Section 6. Other factors involved in obesity progression
- lack of information on the role of ambient temperature, especially the comparison of thermoneutrality vs. standard temperature of 22 °C, in terms of obesogenic effects of various diets, especially those with a high fat content (refere to the thermogenesis paragraph in Section 8)
Section 6. DIT dynamics in obesity
- when discussing how changes in calorie intake (decrease vs increase) trigger a compensatory response in energy expenditure, some classic publications on this topic should be cited and commented on (e.g. Leibel RL et al., New Engl J Med 1995).
Section 11. Conclusions
In order to properly perform obesity studies in laboratory rodents, one of the conclusions of this review should be a recommendation on the publication of complete data on age, strain and substrain of animals used, energy density as well as macronutrient and fatty acid composition of the diets, name of suppliers (animals and diets), etc.
(ii) the comprehensibility and accuracy of the text
- various sentences and expressions need to be corrected, e.g. page 4, lines 157-158 (In contrast,….), p. 5, line 211 (…uncertain fields,…)
- the whole section 8 (i.e. Components of DIT and energy expenditure) needs to be revised due to the presence of incomprehensible formulations (e.g. p. 7, lines 268-271; line 280 (“Energy is the ability to do work”???)
- Section 8,– please, reformulate the last sentence
- Section 10, pge 8, line 17: ….higher caloric intake… = higher energy content
Reviewer 2 Report
Bastías-Pérez and colleagues have written a nice review looking at the factors that need to be considered when designing obesity studies in rodents. The paper is nicely structured, and the illustrations are easily understandable.
My only concern is that the paper is a bit too enthusiastic about rodent models. Variously differences are known between humans and rodents and many findings in rodents have been impossible to replicate in humans. I would therefore like the authors to comment more on the limitations of rodent models and highlight studies with conflicting results between humans and rodents.
results could be replicated in humans. Also if the authors discuss the consequences of animals models being picked by their ability to gain weight and not by their relatability to humans.
Round 2
Reviewer 1 Report
The manuscript has been improved following the comments.